# Implication of Spiritual Network Support System in Epigenomic Modulation and Health Trajectory

**DOI:** 10.3390/ijerph16214123

**Published:** 2019-10-25

**Authors:** Laurens Holmes, Chinacherem Chinaka, Hikma Elmi, Kerti Deepika, Lavisha Pelaez, Michael Enwere, Olumuyiwa T. Akinola, Kirk W. Dabney

**Affiliations:** 1Nemours Children’s Healthcare System, Nemours Office of Health Equity and Inclusion, Wilmington, DE 19803, USA; Chinacherem.Chinaka@nemours.org (C.C.); Hikma.Elmi@nemours.org (H.E.); Keerti.d@yahoo.com (K.D.); Lavisha.Pelaez@nemours.org (L.P.); Michael.Enwere@nemours.org (M.E.); Tolu.Akinola@nemours.org (O.T.A.); Kirk.Dabney@nemours.org (K.W.D.); 2Biological Sciences Department, University of Delaware, Newark, DE 19716, USA; 3College of Population Health, Thomas Jefferson University, Philadelphia, PA 19107, USA; 4Public Health Department, Eastern Virginia Medical School, Norfolk, VA 23507, USA; 5Community and Environmental Health Department, Old Dominion University, Norfolk, VA 23507, USA; 6Public Health Department, Walden University, Minneapolis, MN 55401, USA; 7Sidney Kimmel Medical School, Thomas Jefferson University, Philadelphia, PA 19107, USA

**Keywords:** spiritual network system, epigenomic modulation, gene expression, social adversity, health outcomes, religiosity

## Abstract

With challenges in understanding the multifactorial etiologies of disease and individual treatment effect heterogeneities over the past four decades, much has been acquired on how physical, chemical and social environments affect human health, predisposing certain subpopulations to adverse health outcomes, especially the socio-environmentally disadvantaged (SED). Current translational data on gene and adverse environment interaction have revealed how adverse gene–environment interaction, termed aberrant epigenomic modulation, translates into impaired gene expression via messenger ribonucleic acid (mRNA) dysregulation, reflecting abnormal protein synthesis and hence dysfunctional cellular differentiation and maturation. The environmental influence on gene expression observed in most literature includes physical, chemical, physicochemical and recently social environment. However, data are limited on spiritual or religious environment network support systems, which reflect human psychosocial conditions and gene interaction. With this limited information, we aimed to examine the available data on spiritual activities characterized by prayers and meditation for a possible explanation of the nexus between the spiritual network support system (SNSS) as a component of psychosocial conditions, implicated in social signal transduction, and the gene expression correlate. With the intent to incorporate SNSS in human psychosocial conditions, we assessed the available data on bereavement, loss of spouse, loneliness, social isolation, low socio-economic status (SES), chronic stress, low social status, social adversity (SA) and early life stress (ELS), as surrogates for spiritual support network connectome. Adverse human psychosocial conditions have the tendency for impaired gene expression through an up-regulated conserved transcriptional response to adversity (CTRA) gene expression via social signal transduction, involving the sympathetic nervous system (SNS), beta-adrenergic receptors, the hypothalamus-pituitary-adrenal (HPA) axis and the glucocorticoid response. This review specifically explored CTRA gene expression and the nuclear receptor subfamily 3 group C member 1 (NR3C1) gene, a glucocorticoid receptor gene, in response to stress and the impaired negative feedback, given allostatic overload as a result of prolonged and sustained stress and social isolation as well as the implied social interaction associated with religiosity. While more remains to be investigated on psychosocial and immune cell response and gene expression, current data on human models do implicate appropriate gene expression via the CTRA and NR3C1 gene in the SNSS as observed in meditation, yoga and thai-chi, implicated in malignant neoplasm remission. However, prospective epigenomic studies in this context are required in the disease causal pathway, prognosis and survival, as well as cautious optimism in the application of these findings in clinical and public health settings, due to unmeasured and potential confoundings implicated in these correlations.

## 1. Introduction

While human social conditions have been largely implicated in disease process and prognosis, the biological implications of these social impacts, namely social isolation, low socio-economic status (SES) and unstable social status, have been investigated in animal and human models. Epidemiologic and experimental data observed a common pattern characterized by increased expression of conserved transcriptional response to the adversity (CTRA) gene in social isolation, low SES, unstable social status and chronic stress [1]. Specifically, these conditions elicit social signal transduction via sympathetic nervous system and the beta-adrenergic receptor activation [2,3]. In addition, prolonged and chronic stress have been implicated in sustained activation of the pituitary and hypothalamus brain region, thus diminishing the negative feedback mechanism involved in the allostatic response to stress via cortisol elaboration. The down-regulation or decreased expression of the glucocorticoid receptor gene, namely NR3C1, has been widely observed in early life stress (ELS) or early life adversity (ELA), which lead to psychopathologic conditions such as major depressive disorders in adolescence and adulthood [4].

The mediating biologic consequences of maternal deprivation and lack of care of offspring in animal models clearly illustrate impaired gene and environment interaction as indicated by aberrant epigenomic modulation of the candidate genes involved in these conditions [5,6,7,8]. Similarly, low SES, which reflects income inequalities, implies variabilities in access to social, educational, health and other resources, resulting in anxiety, depression, shame, self-harm and other psychopathologies [9,10,11,12]. Data on social hierarchies driven by high SES observed privileged access to social determinants of health (SDH) and basic life resources such as water, food and shelter as appropriate living conditions. The SES which correlated with anxiety in most cases as observed in capitalistic society reflects social adversity as an exposure function of childhood physical and emotional neglect, restricted control over decision-making and life choices as well as marginalized social interaction and decreased trust and confidence with societal members [13]. Epidemiologic and clinical data indicate hyepercortisolemia and increased pro-inflammatory cytokines in low social ranked individuals as a marker of social gradient in health [9]. Available animal and human model studies observed DNA hyper-methylation of the CpG regions of the genes, such as NR3C1, involved in allostatic stress response. However, it is not fully understood if the neural activities involving the hypothalamus pituitary adrenal (HPA) axis and the cortisol-mediated negative feedback mechanism induced by social signal transduction via social isolation, low SES, unstable social status and ELS, such as child abuse and neglect, is regulated solely by NR3C1.

While CTRA has been observed with respect to individual happiness, termed hedonic happiness or self-satisfaction, the individual happiness resulted from noble course, also termed auto-transcendence, present with distinct levels of the related gene expression and correlated health outcomes. Social network implying enhanced parenting skills, family interactions and individual capacity development has been shown to improve health outcomes, while social network disruption induces low SES and poor health outcomes, as well as decreased proinflammatory cytokines [14,15]. Previous studies observed how the DNA methylation profile differentiates these types of happiness, resulting in improved health outcomes with auto-transcendence driven by altruism [16]. With the observed data on differential DNA methylation and the correlated health outcomes and the observation of a spiritual network support system (SNSS) as a positive social interaction and antithesis to social isolation and unstable social status, there is a need to examine how SNSS enhances human gene expression, health status, prognosis and survival. The understanding of how social conditions such as spiritual support network system influences the biologic process in pathophysiologic mechanism of disease causation and prognosis allows for specific risk characterization and reliable intervention mapping with biomarkers identification and social policies implementation. We aimed in the current study to relate SNSS with biological alteration as human social condition and to correlate the attributes of this subpopulation with the DNA methylation of the human genes involved in stress and social isolation. In addition, this review suggests the design of original studies in this direction and potential policy formulation in stress reduction and human health improvement via religious diversity and autonomy.

## 2. Materials and Methods

### 2.1. Study Design

A systematic review, implying the study of studies, was used to examine published literature on the potential or implicit role of the spiritual support network system in epigenomic modulation and health outcomes. This approach is feasible, given limited or sparse data on causal inference in SNSS and gene interaction.

### 2.2. Search Strategy and Eligibility Criteria

The Medline, EMBASE and the Cochrane library were utilized to identify published literature on social stress, social human conditions and gene interaction as epigenomic modulation investigation. Additionally, the health outcomes of these modulations were examined. The combination of these search terms yielded several articles, but very few studies were identified with the spirituality and biomarkers of disease and health outcomes. The search terms used were: “stress and gene expression”, “social stress and gene expression”, “early life stress and epigenomic changes”, “chronic stress and DNA methylation”, “NR3C1 gene and social adversity”, “social threats, isolation and CTRA gene expression” and “social adversity, epigenomics and inflammatory response”.

### 2.3. Data Quality

Studies involved in the qualitative systematic review were assessed for their: (a) Objective and reliable research questions, (b) outcome and predictor variables measures, (c) design, (d) sample size and power estimations, (e) confounding and effect measure modifier, (f) bias minimization and random error quantification, (g) statistical analysis, (h) follow-up and attrition rate, (i) measure of effect and point estimate, as well as precision estimate for the parameter estimation (effect size or point estimate).

### 2.4. Study Synopsis

The studies utilized in the result section were examined for their adequacy in terms of the background, materials and methods, data interpretation, discussion and inference. A common theme and direction of the findings were identified and assessed for reliability based on critical appraisal of literature prior to incorporation into the review and the scientific statement as inference in the current review.

## 3. Results

### 3.1. Results

#### 3.1.1. Human Biologic System and Complex Social Interaction

Human beings are meta-organisms, implying a complex social and physical ecosystem in survival and adaptation. The survival of humanity depends on how best this meta-organism adapts to everyday changes in the human environment, which has been shown to depend on social, economic, chemical and physical conditions related to normal physiology and health. While humans are the products of inheritance or genetic composition of parents, implying that traits are carried from each of the parents, based on the gene that evolves from the twenty-three pairs of chromosomes, human adaptation in terms of response to the environment depends on how these pairs of chromosomes communicate or interact with these social, economic, chemical and physical environments. Specifically, 23 pairs of chromosomes which contain several genes as molecules or chemical substances required for several cells’ development and maturation are inherited from parents at gametogenesis (conception). With each pair from the father (sperm cell) and mother (egg cell) combining to form 23 pairs, humans are composed of chromosomes, which contain genes and genes are made up of deoxyribonucleic acid (DNA). The information needed to make various cells for the organ systems require the order or the sequencing of the DNA based on the dinucleotide paring, namely adenine (A), cytosine (C) thymine (T) and guanine (G) in this order, C-G and T-A. This ongoing interaction is termed epigenomic modulations, commences from gametogenesis and is reversible but transgenerational. For example, 48 hours’ intrapartum stress may result in early life stress (ELS), predisposing an adolescent to a major depressive episode (MDD), through adverse epigenomic modulation with DNA methylome impairing the glucocorticoid receptor gene (NR3C1) expression.

#### 3.1.2. Physical and Chemical Environment and Gene Interaction

Over the years, changes in human health have been observed to correlate with adverse physical and chemical environments, such as residing close to gas stations, landfills and incinerators, adversely affecting human and animal health and resulting in increased mortality [17]. This adverse neighborhood environment factor, which is predominantly associated with low SES, mainly among black/African Americans (AA), inversely correlates with their health outcomes [17]. The environmental effects of air pollutants, toxic waste and polluted water have been linked with subpopulation differences in asthma, respiratory diseases and cancer [17,18]. This observation reflects how the exogenous environment influences endogenous environment, such as hormones and predisposes to disease outcome among socially disadvantaged individuals. Moreover, we have observed how genetic conditions affect human health and result in a short life span. While a gene is implicated in disease development, the subpopulation differences in disease, prognosis and survival is not due mainly to subpopulations’ genetic heterogeneity, but variability in social determinants of health (SDH) such as living conditions, geocoding and the interaction of these attributes with the biologic system, mainly genes, in determining subpopulation differences in morbidity and mortality [19].

#### 3.1.3. Social Conditions and Gene Expression

While gene and physiochemical interactions and disease outcome observations remain valid and accurate, we are beginning to experience more than ever before the role played by social isolation, low socio-economic status, discrimination, unstable social hierarchies, repeated social threats and unstable social status, which results in chronic stress, depression, chronic diseases, cardiovascular disease and cancers [20,21,22,23,24]. In effect, social environments comparing individuals isolated versus those not isolated have been shown to result in differences in the pro-inflammatory gene expression, such as IL6, antibodies (immunoglobulins) and interferon-gamma. Specifically, repeated social threats or social isolation induces increased expression of pro-inflammatory genes resulting in increased production of pro-inflammatory cytokines, namely IL-6, while inhibiting the gene expression in antibodies synthesis (response to pathogenic microbes) and interferon gamma (innate response to viral pathogens). This observation very clearly indicates decreased elaboration of antibodies synthesis, such as immunoglobulin G (IgG) and increased elaboration of interferon-gamma (IFN-γ) as a result of the increased response of the conserved transcriptional response to the adversity (CTRA) gene. When comparing those isolated with those not isolated, available studies have indicated that there was an estimated 5% difference in the specific genes involved with this condition [25]. However, in terms of the gene and environment, more than a 50% difference has been observed comparing individuals isolated versus those not isolated [25]. These studies very clearly affirm the role played by genes and environments in disease development.

Consequently, it is not the gene per se that indicates the differences in the outcome of isolation, but the gene in environment interactions, termed epigenomic. Specifically, social conditions play a substantial role in human gene expression, which has been previously observed in animal models. Simply, stress or isolation evokes the sympathetic nervous system response (SNS) within the central nervous system (CNS) leading to increased expression of the CTRA gene. This increased response of the CTRA gene has been observed in leukocytes due to repeated social threats, unstable social hierarchies and low social status [26]. The observed SNS activation of CTRA gene up-regulation is due to the beta-adrenergic receptor response, leading to the elaboration of some transcription factors such as nuclear factor kappa-light chain-enhancer of activated B-cell (NF-kB) and CAMP response element-binding (CREB) proteins. These selective cascade results in increased pro-inflammatory gene expression such as interleukin 6 (IL-6), IL-1A, IL-1B, tumor necrosis factor (TNF) and down-regulation of the transcription factor responsible for IFN-γ gene expression.

Additionally, the CTRA is involved in transcriptional regulation of the human immune system cells via the transcriptional shifts in the myeloid cells, namely monocytes and dendritic cells. Simply, the SNS response to stress or social isolation can up-regulate pro-inflammatory monocytes elaboration via alteration of the hematopoietic process in the bone marrow. Furthermore, beta-adrenergic signaling has been shown to up-regulate the transcription of the myelopoietic growth factor granulocyte-macrophage colony-stimulating factor, influencing monocyte development, differentiation and maturation [27]. In effect, social threats or adversity have potential in up-regulating CTRA gene expression in the human immune system, leading to chronic inflammatory response and observed excess malignant neoplasm, diabetes and chronic disease [3,28]. Substantial data supports the implication between psychological, neural and endocrine processes in cancer patients and gene expression; however, cautious interpretation of these data in avoiding reverse causation are required since malignant neoplasm may induce inflammatory processes with subsequent effects on CNS, SNS and beta-adrenergic receptor-mediated social signal transduction and impaired gene expression.

#### 3.1.4. Mechanistic Process in Gene Expression: DNA Methylation

A genomic and epigenomic process that involves a specific gene expression reflects the inherent ability of humans to respond to sporadic and transient threats, creating cellular plasticity. For instance, the conserved transcriptional response to adversity (CTRA) gene expression occurs as a result of a stressful environment such as objective and subjective isolation and results in a molecular level adaptive process. However, when the stressful circumstances become chronic and sustained over time, there emerges a pro-inflammatory response with the potential for type 2 diabetes, atherosclerosis, hypertension, neurodegeneration, malignant neoplasm, etc. Additionally, the sustained expression of a CTRA gene compromises the immune responsiveness by antibody synthesis inhibition (IgG), as well as decreased elaboration of IFN-γ, as an innate response to viral pathogens.

To understand epigenomics, implying the interaction between the social, spiritual, economic and physical environment such as toxins and pollutants, there is a need to understand the specifics of hereditary material as we evolve as humans; mainly genes. The hereditary material of a gene comprises a deoxyribonucleic acid (DNA) molecule, which is present in all cells in the human body and resembles a coiled ladder. The scientifically termed double helix contains all the information that constitutes individual characteristics. The DNA consists of four nucleotides, namely adenine, cytosine, guanine and thymine, a phosphate and a sugar molecule. The combination of these nucleotides forms the codes that are required for the synthesis of amino acids, which are the building blocks of protein molecules. The process upon which the protein molecules are developed for continued molecular and cellular functioning is based on the dogma of molecular biology that begins with the replication of the DNA, transcription of DNA into RNA and the subsequent translation of the messenger RNA into protein synthesis—Replication (R) → Transcription (T) → Translation (T).

Humans are compositions of cells (brain cells, muscle cells, bone cells, reproductive cells, skin cells, endocrine cells) with these specialized cells powered by genes (cell roadmaps), in terms of development, differentiation and maturation. Proteins that are required for cellular functions are synthesized through the RTT process commencing from DNA, the genetic material. The DNA replication involves the uncoiling of the double helix with the complementary base pairing nucleotides (G-A, C-T) leading to the transcriptional activities, requiring the translation of the codes by mRNA into amino acids, the building blocks of protein. In effect, these steps require very tight regulation and any error may result in damage or inhibition in the process resulting in impaired protein synthesis and the loss of cellular function, which may lead to poor tissue modeling and impaired organ-system integration. 

The interaction between the gene and indigenous, exogenous, social, psychological and environmental factors is an ongoing process occurring every moment and every instant of human existence. This process is termed epigenomic modulation, meaning above the gene, and reflects changes outside the coding region of the gene, but does not involve the DNA sequence. The normal changes or modulation in this direction reflect the stability of the genome and the normal regulation of everyday activities and health outcomes. In contrast, when there is an insult in this modulation, there occurs poor health outcomes, including increased incidence of disease, poor prognostic and excess mortality. Specifically, epigenomic modulation begins with cell signal transduction that involves the transcriptome factors and the subsequent utilization of these factors, which are protein molecules in gene expression. The epigenomic mechanistic process has been observed to affect DNA methylation, histone acetylation (ACH), hydroxyl-methylation (OH-CH3) and other mechanisms not required in this basic explanation. However, of this mechanistic process, the most fundamentally utilized is DNA methylation. This process begins at the promoter region of the gene, or the gene enhancer region, which is populated with cytosine and guanine residue (CpG island). While normal epigenomic modulation involves the DNA methyltransferase recruitment of the CH3 molecules to the CpG island, aberrant epigenomic modulation involves dense DNA methylation, also termed DNA hyper-methylation, that results in the inhibition of the transcriptome and the subsequent impairment in the gene expression, adversely affecting protein synthesis and cellular function. Consequently, the methylation of the cytosine residue, resulting in dense DNA methylation at the CpG region, influences gene expression, inducing cellular dysfunctionality.

#### 3.1.5. Social Signal Transduction and Gene Expression

The available literature on social isolation or stress, as an associated social signal transduction (SST) with changes in the sympathetic nervous system (SNS), clearly indicates how stress signals the fight or flight emotions and reactions that eventually results in alterations that affect how the gene expresses itself. Specifically, the involvement of the sympathetic nervous system results in the elaboration or production of the beta adrenergic response and the subsequent increase in the receptors associated with this elaboration. Social contexts involving stress, loneliness and social threats include the tendency to provoke the HPA axis response, resulting in cortisol elaboration in response to stressful conditions. This response, which involves the glucocorticoid receptor gene, plays a protective function, enhancing inflammatory recovery. However, chronic stress has been shown to up-regulate this response, leading to chronic diseases, metastatic cancer and metabolic syndromes, indicative of the loss of cellular plasticity and the subsequent cellular damage and delayed response to damage, impairing cellular repair and restoration [25,29].

The observed CTRA gene expression has a role in protecting humans from transient and short episodes of threats, social instability and low SES. However, sustained and socially organized and structured social isolation, chronic stress and discrimination result in the loss of this adaptive molecular mechanism, impairing cellular plasticity, with such impairment reflecting epigenomic lesion or aberration. The obvious questions remain in characterizing specific social contexts that inhibit CTRA gene expression following social adversity, thus enhancing transcriptional activities related to quality health and normal molecular level events. Another relevant issue in the social epigenomic remains, that of quantifying and categorizing spiritual network interaction with human gene expression or mRNA sequence. The multi-dimensional nature of spirituality or religious affiliation renders the nexus between worshiping as a supportive social network and gene expression a challenging notion. With this challenge, data have observed the distinction between hedonic happiness as self-gratification or auto-focus and eudemonic happiness as auto-transcendence or commitment to a noble course or purpose in enhancing the happiness and self-realization of others, outside one’s self. Of these forms of happiness, where do we categorize generic worshiping characterized as meditation? Available data observed decreased expression of the CTRA gene among individuals with eudemonic happiness relative to hedonic happiness [16]. The health effect of such observation signals a significant reduction in pro-inflammatory cytokines, such as IL-6, and increased antibodies production and IFN-γ as an innate antiviral response. Invariably, eudemonic happiness as a surrogate of spirituality enhances normal epigenomic modulation, improving human health. With the different dimensions of spiritual wellbeing or affiliation, a review of Christian, Muslim and Hindu patterns of worshiping observed the following activities.

### 3.2. Spiritual Network System and Gene Expression: Normal Epigenomic Modulation

A spiritual network system reflects the organized structure by individuals or groups with the notion of transcendence based on hope, faith and divine providence, indicative of a supernatural aspiration. This concept is perceived as a social condition and plays an important role in epigenomic modulations. However, this impact may vary depending on the activities that define spiritual network systems as either interpersonal or intrapersonal social support network systems. Since reliable data have implicated epigenomic modulations associated with social conditions in animal survival and animal health [30], a spiritual network system might portray similar input, requiring more prospective investigations on the health benefits of socio-epigenomic related to SNSS.

Neurophysiological alterations have been linked with social stress due to social environment interruption. Studies on social stressors clearly implicate the epigenomic mechanism in the translation of social stressors such as isolation and a state of psychosocial wellbeing in biologic and physiologic adaptation. Specifically, available data on early life stress (ELS), due to parenting deficits at infancy and childhood, inversely correlate with physiologic stability and normal health [31]. In addition, low SES, psychological trauma and social isolation with prolonged or sustained exposure result in aberrant epigenomic modulation and the related adverse health outcomes. While in animals and humans, an allostatic mechanism is required in response to acute and transient stress. Prolonged stress impairs this adaptive mechanism, resulting in allostatic overload and hence ineffective adaptation, physiologic alteration and poor health outcomes [32]. The observed epigenetic mediation of stress is explained in part by human neural circuits that perceive, process and react to psychosocial threats or stressors. The biologic or epigenomic implication in response to chronic stress is illustrated in the transgenerational nature of epigenomic signatures, where animal and human models show the health or physiologic consequence of exposure to chronic stress in offspring of those exposed, suggestive of aberrant epigenomic signatures’ transmissibility.

An example of social stressors as ELS has been illustrated in pre- and postnatal stress and maternal care insufficiencies associated with behavioral pathologies [33,34,35]. Animal models with rodents demonstrated an inverse correlation between adequate postnatal maternal care and increased stress reactivity, anxiety and fear reaction [7]. In general, the main neural substrates involved in the epigenetic modulation of ELS and maternal care include HPA, amygdala, medial prefrontal cortex and the hippocampal region of the brain. The observed phenotypes in the aberrant epigenomic modulation in ELS is explained by the NR3C1 gene down-regulation and decreased expression, a glucocorticoid receptor (GR) gene. The animal model in this context observed the dense methylation of the CpG in the promoter region of NR3C1, increased activity of the HPA axis and elevated serum glucocorticoid level [7,8,36]. Additionally, the animal model reveals increased stress reactivity, anxiety, impaired social interaction and depressive episodes associated with maternal separation. The observed manifestations correlated with genome-wide aberrant DNA methylation and brain neural pathway gene transcripts [37,38,39].

A prolonged activation of the HPA axis, altered gene transcription and aberrant DNA methylation were observed in CNS and peripheral T lymphocytes among infants and juveniles deprived of maternal care [40,41,42,43,44,45]. Aberrant epigenomic modulation (mDNA) and hydroxymethyl cytosine at the brain-derived neurotrophic factor (BDNF) locus, a member of the nerve growth factor family of neutrophils, has been observed in adolescence and adulthood distress associated with maternal maltreatment during neonatal nursing of offspring. Other neural changes include hippocampus, amygdala and medial prefrontal cortex [46,47,48,49]. Further studies observed increased the expression of arginine vasopressin (Avp) and gene loci, a protein coding gene for the antidiuretic hormone (vasopressin) associated with diabetes insipidus and DNA hypo-methylation in the paraventricular nucleus of the hippocampus in ELS, due to maternal separation. Similarly, the overexpression of proopiomelanocortin (pomc) gene loci, which is involved in the production of the adrenocorticotropic hormone (ACTH) and binds to the melanocortin 2 receptor (MC2R), stimulating cortisol release, is observed in ELS associated with maternal separation [50]. The overexpression of NR3C1 inversely correlates with corticotrophin-releasing hormone (CRH), transcription in chronic stress induced by ELS [51].

Table 1 illustrates the candidate genes associated with social adversity and the expression inherent in the interaction as well as the biology and health outcomes of such interaction. Human models have clearly illustrated how prenatal, perinatal, infant and adolescence social adversity exposure lead to aberrant epigenomic modulations that are transformational into adulthood. While NR3C1 is implicated in ELS in rodents, varieties of NR3CI sequences are observed as regulatory targets with the most consistent and common epigenomic modulation being hyper-methylation at exon 1F of the human NR3C1 gene in ELS or early life social adversity. The DNA sequence element that encodes the methylation-sensitive binding site associated with the neural activity-regulated transcription activator (NGFTA) is embedded in the exon 1F region of the NR3C1. The hyper-methylation of NR3C1 in ELS, implying prolonged stress that results in allostatic load, interrupts the glucocorticoids mediated to the pituitary and hypothalamus, enhancing HPA axis persistent activation and the consequent psychopathology including chronic inflammation and major depressive disorders (MDD) [52]. A correlation between parental stress such as domestic violence, intimate partner violence and NR3C1 CpG hypermethylation has been observed. Likewise, child abuse and neglect has been shown to be associated with dense NR3C1 CpG methylation and the subsequent impaired NR3C1 transcription, leading to NR3C1 down-regulation and decreased gene expression [53,54,55,56,57,58]. While NR3C1 hyper-methylation results in NR3C1 reduced transcription, it is not clear if other pathways or epigenomic processes influence NR3C1 gene expression. However, noncoding RNA, namely IncRNA, GASS and miRNA, have been shown to impair NR3C1 gene expression and protein activities [59,60].

Other epigenomic modulations associated with early life adversities in animal models have implicated similar genes in humans such as BDNF, CRH, PM2D1 and CHRBP [45,61]. Additionally, genes involved in early life adversities include MAOA, SLC17A3, MORC1 and FKBP5 [62,63,64,65,66,67]. Overall, child maltreatment implying abuse, neglect and parental care deprivation signal aberrant epigenomic modulation inducing psychopathology and chronic disease such as type II diabetes and hypertension in adolescence and adult life due to allostatic overload and marginalized glucocorticoid-mediated negative feedback to pituitary and hypothalamus.

Table 2 demonstrates the gene expressions that are implied in behaviors or activities associated with religious beliefs and worshiping that are contrary to social adversity that results in impaired gene expression. Despite sparse data in epigenomic modulation in Christian, Islamic and Hindu religions, the observed behavior in these religions reflects the potentials for the DNA methylation process that may signal appropriate gene expression, improve cellular activities and optimize health outcomes.

#### 3.2.1. Christian Religious Practice

Christianity, as a culture, involves worshiping, implying performing what is written in the Bible, the Holy Book of God, which is truth, through the Holy Spirit—the inner being of an individual. In context, worshipers are expected to have full expression, connect with their hearts, emotions and mind and be empowered to take action through their individual spirit. The place of worship is characterized by a peaceful environment and tranquility as well as defined by holiness and divinity. While auto-transcendence is the building block of worshiping, within the Christian context, the act of devotion or meditation involves the engagement of the muscles (local motor activities, facial expression and coordination) from dancing and singing which is reflective of physical activities among Christian believers in the act of worship. The notion of God in the Christian religion as omnipotent allows for the Christians’ perception of hope and faith in their everyday interactions in the world. This hope and faith remain a motivation factor for a reduction in risky behavior, as reflected in the Holy Bible, implying the Ten Commandments. In most Christian religion, the notion of communion, implying the “breaking of bread,” clearly illustrates healthy behavior among worshipers, which may contribute to the appropriate gene expression, protective against morbidity and excess mortality in this subpopulation.

#### 3.2.2. Muslim Religious Practice

Of the five pillars of Islam is prayer. Muslims are expected to pray the five following prayers daily: Fajr (the dawn prayer), Dhuhr (the early afternoon prayer), Asr (the late afternoon prayer), Maghrib (the sunset prayer) and Isha’a (the night prayer). Prayer, also known as salah, must be performed in such a way that requires various and repetitive full body movements. The entire body is completely engaged in prayer, which involves standing, bowing, prostrating, kneeling and sitting. These movements have been compared to other ritualistic activities like yoga that promote cardio and circulatory health, increased flexibility, as well as improved respiration, energy and vitality. Physical similarities between salah and yoga are in the body movements that are repeated in a set pattern [68]. The duration of time for each prayer varies depending on the specific prayer and the individual. Salah is considered a form of meditation and it requires the individual to momentarily disengage themselves from the outside world in order to immerse themselves into their spirituality. It ultimately allows the individual to disconnect from their worldly stressors and to better focus on their divine wellbeing. Before an individual begins each salah, he/she must perform ablution (also known as wudu) which is a ritual purification that involves the washing of particular body parts, namely the hands, mouth, nostrils, face, arms, head and feet—in that order—with water, to remove physical impurities. The complete act of salah (including wudu) fundamentally cleanses the individual both externally and internally often to the point that once the individual has completed the salah, they are essentially in a better mental and physical state. Studies have shown that regular salah promotes relaxation, minimizes anxiety and reduces cardiovascular risk because parasympathetic activity is increased and sympathetic activity is decreased [69].

#### 3.2.3. Hindu Religious Practice

Many practices are prevalent in Hinduism in the name of meditation. Meditation purifies the mind from various impressions created through action and reaction. In the purified state of mind, through self-enquiry, self-realization can take place, which is the goal of life in Hinduism. Hence the importance of meditation in Hinduism. In meditation, one becomes a neutral observer of all thoughts as they come and go and soon the mind falls silent. It is not an experience because in this state the experiencer and the experience become one, there is only oneness. A calmer, relaxed mind after an intense yoga session seems to go hand in hand with bolstered health and a reduced chance of disease. Yoga has long been associated with increasing wellbeing and reducing stress. Since inflammation has been connected to stress and depression, engaging in physical activities like yoga and meditation, which reduce inflammation and promote anti-inflammatory factors, can be very beneficial. A noticeable decrease in IL-6 is found in women who practice yoga as it has a well-known role in the pro-inflammatory processes, but is increasingly recognized in healing and regeneration activities.

## 4. Discussion

Overall, these religious activities, while reflecting eudemonic happiness, also portray increased physical activity relative to individuals with sedentary lifestyles implying potentials for down-regulation of the CRTA gene, as well as allostatic response to stress mediated by NR3C1 gene expression, involved in cortisol response to stress and the HPA axis activities. The investigation of reliable and valid gene expression or epigenomic modulation implicated in spirituality will require specimen collection from the exposed and non-exposed subjects for evidence of DNA methylation of the promoter regions of the implicated genes, namely NR3C1, for normal epigenomic modulation in those exposed to prayer, and meditation relative to the non-exposed Muslims, Hindus or Christians.

The implication of stress in chronic disease and cancer prognosis has been shown to correlate with conserved transcriptional response to adversity (CTRA) gene expression, which reflects pro-inflammatory response as observed in leukocytes (white blood cells). Studies have indicated that cognitive behavioral stress management, meditation, yoga, Thai-chi and exercise improve the outcome of cancer in patients diagnosed with malignancies and treated for the disease [52,53,70,71,72,73,74,75]. The observed improvement in cancer outcomes is associated with the down-regulation of the CTRA gene expression, which enhances the transcriptome due to the normal methylation at the CpG region of the gene since dense DNA methylation enhances aberrant epigenomic modulation.

The social network support system includes interpersonal relationships, family interaction, community interaction, spirituality, religious affiliation, prayers and divine interaction. The individual involved in these networks correlates with healthy gene expression and favorable health outcomes, including morbidity reduction, good prognosis, survival advantage and lower rates of mortality. The observed gene expression associated with spirituality and religious activities as a function of network systems is explained in part by individual and altruistic happiness as well as avoidance of risky health behaviors, such as excessive alcohol consumption, drug use/abuse and risk-taking behaviors. Within the worshiping context, the church or synagogue represents both the vertical and horizontal projection in transcendence, implying an individual’s belief in the divine and the application of such belief in improving an individual’s lifestyle.

Early studies on gene expression observed significant variances between gene and environmental input on gene expression in leukocytes which are the white blood cells that are up-regulated following exposure to pathogenic microbes and in chronic inflammation. Specifically, household poor air quality had been implicated in human health which is explained in part by aberrant epigenomic modulations [76]. Specifically, as observed earlier, an estimated 5% was observed in traits loci relative to more than 50% differences in gene expression based on differences in urban and rural social environments [25].

Using spirituality of worshiping as an antithesis to social isolation, stress, low social economic status (SES), unstable social status and racial discrimination, it is plausible to consider social signal transduction in this context to be associated with parasympathetic response, resulting in the inhibition of the beta-adrenergic receptors and decreased expression of the CTRA gene. This observation signals increased elaboration of antibodies and the inhibition of IFN-γ, leading to good health outcomes and decreased disease development.

## 5. Future Perspectives and Health Implications

In effect, given the role of social network systems in human health and the implication of religious and spiritual activities in this context, there is an inclination to consider how spiritual activities could improve human gene expression due to normal epigenomic modulation via DNA methylation processes involved in CTRA and NR3C1 gene expressions. However, such inclination or observation should be cautiously interpreted, given that individuals who are involved in worshiping or religious activities are less likely to consume alcohol excessively, and be involved with drug use and abuse as well as risky behaviors in adverse human health. Therefore, future studies intended to address the role of spirituality in human gene expression must control for the social determinants of health before reliable findings. The availability of these data should facilitate intervention development by advocating SNSS early in human growth and development in reducing stress as well as chronic stress reversal in some populations at risk, especially the socially disadvantaged individuals affected by the social gradient in disease development, prognosis and survival. Additionally, available data on normal epigenomic modulations should allow for policy development in advocating religious diversity in schools and communities for religious autonomy and SNSS for health equity transformation in all cultures

## 6. Conclusions

In summary, a spiritual network support system (SNSS) as a human psychosocial condition enhances social interaction resulting in normal gene expression via down-regulation of CTRA and appropriate expression of the NR3C1 gene, resulting in normal health and decreased adverse health outcomes. The SNSS as a positive social network may enhance adequate allostatic response to stress through normal epigenomic modulation reducing pro-inflammatory cytokines and improving human health. These review findings are suggestive of the need to conduct prospective studies in SNSS gene interaction associated with neurobehavioral disorders, malignant neoplasm, cardiovascular disease and metabolic syndromes. Additionally, these data are indicative of the public health and clinical medicine needed to advocate adequate SNSS through social policies and intervention mapping in stress reduction via prayers and meditations, thus ameliorating human health.

## Figures and Tables

**Table 1 ijerph-16-04123-t001:** Candidate genes implicated in social adversity and health outcomes.

Social Adversities	Gene and Epigenomic Mechanism	Biologic Mediation and Health Outcomes
Social Isolation Social Threats	CTRA up-regulation	Synthesis of antibodies and impaired elaboration of interferon gamma (IFN- γ) Pro-inflammatory cytokine (IL-6, IL-1A, IL-1B) NF-KB TNF Type II diabetes, neuro-degenerative conditions, arteriosclerosis, malignant neoplasm
Low Socio-economic Status (SES)	CTRA up-regulation	Type II diabetes, neuro-degenerative conditions, arteriosclerosis, malignant neoplasm
Maternal Separation	NR3C1-glucocorticoid receptor gene down-regulation and impaired hypothalamus-pituitary-adrenal (HPA) axis	Impaired glucocorticoid-mediated negative feedback mechanism resulting in allostatic overload Chronic inflammation, major depressive disorder (MDD)
Early Life Stress	NR3C1 hyper-methylation at exon 1F promoter region, down-regulation and impaired gene expression BDNF gene methylation (down-regulation) CRH gene methylation (down-regulation)	Major depressive disorder, suicide, chronic inflammatory disease

**Notes and abbreviations**: CTRA gene refers to the conserved transcriptional response to the adversity gene, which is up-regulated under stress mediated through leucocytes. The BDNF gene, which is the brain-derived neurotropic factor, is a growth factor family of neutrophils, an immune cell involved in nonadaptive and nonspecific immune responses.

**Table 2 ijerph-16-04123-t002:** Candidate genes implicated in social interaction and implied spiritual network support system (SNSS).

Social Support Experience	Gene and Epigenomic Mechanism	Biologic Mediation and Health Outcomes
Devotion Meditation	CTRA regulation	Synthesis of antibodies and impaired elaboration of interferon gamma (IFN-γ) Pro-inflammatory cytokine (IL-6, IL-1A, IL-1B) NF-KB TNF Incidence of diabetes, neuro-degenerative conditions, arteriosclerosis and malignant neoplasm
Yoga Thai-chi	CTRA down-regulation	Incidence of diabetes, neuro-degenerative conditions, arteriosclerosis and malignant neoplasm
Cognitive behavior stress Management	CTRA up-regulation Normal sympathetic nervous system response Normal beta-adrenergic receptor response to social signals	Normal glucocorticoid-mediated negative feedback mechanism resulting in adequate allostatic response Inflammation
Prayers and activities associated with worshiping	NR3C1 hypo-methylation at exon 1F promoter region, up-regulation and normal gene expression Normal expression of CHR, BDNF, ACTH	Major depressive disorder, chronic inflammatory disease, cardiovascular disease, child maltreatment and increased parental care

**Notes and abbreviations**: CTRA gene refers to the conserved transcriptional response to adversity gene, which is up-regulated under stress mediated through leucocytes. The BDNF gene, which is the brain-derived neurotropic factor, is a growth factor family of neutrophils, an immune cell involved in nonadaptive and nonspecific immune responses.

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
