# Peer review of "Implication of Spiritual Network Support System in Epigenomic Modulation and Health Trajectory"

_ijerph, 2019, doi:10.3390/ijerph16214123_

Round 1

Reviewer 1 Report

Well written, well organised and thorough.The authors are to be congratulated on their diligence .

The paper, though sound overall, could have had better balance if the links to the Spiritual Support System were explored and discussed in more depth.The epigenomic processes is where the authors clearly felt on more secure ground. The abstract could have been more succinct and sharply focussed. The conclusion could have been stronger.

The systematic review is well organised and has depth. A conclusion maybe that more investigation in the connection between spirituality and health and the implications for public health, is an appropriate follow on (there is a deal of research on this to start from)

A minor revision to tighten the abstract and strengthen the conclusion would help the paper achieve more.

Author Response

Reviewer comments:

Well written, well organised and thorough.The authors are to be congratulated on their diligence.

Author’s response: Thank you very much for the observation. We have carefully reviewed the abstract and we have addressed it accordingly.

Reviewers comment: The paper, though sound overall, could have had better balance if the links to the Spiritual Support System were explored and discussed in more depth. The epigenomic processes is where the authors clearly felt on more secure ground. The abstract could have been more succinct and sharply focussed. The conclusion could have been stronger.

Author’s response: Thank you for the comment regarding the abstract clarity. We have reexamined the abstract and identified the limitations which resulted in the enhanced clarity and readability. Once again, thank you for the privilege for us to enhance the quality and reliability of this information for application. We also addressed the conclusion by stating the findings and suggesting further prospective studies to be done in identifying the specific gene expressions associated with social interaction as a component of spiritual network support system.  

Reviewers comment: The systematic review is well organised and has depth. A conclusion maybe that more investigation in the connection between spirituality and health and the implications for public health, is an appropriate follow on (there is a deal of research on this to start from)

Author’s response: Thank you very much for the comment and observation. We have enhanced the conclusion by clearly indicating the findings from the reviewed literature as well as provide suggestion for prospective studies in these direction. Please see the bold (conclusion).

Reviewers comment: A minor revision to tighten the abstract and strengthen the conclusion would help the paper achieve more.

Author’s response: Once again, we are appreciative of this comment and suggestion of yours and have carefully addressed the abstract and drawn a reliable conclusion based on based on the reviewed papers in this study.

Reviewer 2 Report

The Material and Method section should be extended. It is necessary to describe the used techniques more detail. The obtained results also should be justified more clearly. The type of thearticle is not clear. If this is rewiew, the structure should be other. If it is science paper, is where the authors' research results?

Author Response

Reviewer comments: The Material and Method section should be extended. It is necessary to describe the used techniques more detail. The obtained results also should be justified more clearly. The type of the article is not clear. If this is rewiew, the structure should be other. If it is science paper, is where the authors' research results?

Author’s response: Thank you very much for the observation and the comment regarding the manuscript. We have carefully assessed your comment and find it inapplicable to review articles or papers. Prior to your comment, we had applied the journal requirements in preparing the method section of this review article. In addition, the results of a review paper is nothing but a synthesis of the original articles based on the research questions required by the review article. This, we have clearly addressed. As a review paper, there is no requirement for the result section involving specific output interpretation supported by data. Therefore, these comments are irrelevant to review papers. Once again, thank you for your comments and suggestions.

Round 2

Reviewer 2 Report

I have not any questions